# Prevalence of HPV and Assessing Type-Specific HPV Testing in Cervical High-Grade Squamous Intraepithelial Lesions in Poland

**DOI:** 10.3390/pathogens12020350

**Published:** 2023-02-19

**Authors:** Marcin Przybylski, Dominik Pruski, Katarzyna Wszołek, Mateusz de Mezer, Jakub Żurawski, Robert Jach, Sonja Millert-Kalińska

**Affiliations:** 1Gynecology Specialised Practise, 60-682 Poznań, Poland; 2Department of Obstetrics and Gynecology, District Public Hospital in Poznan, 60-479 Poznań, Poland; 3Gynecology Specialised Practise, 60-408 Poznań, Poland; 4Department of Maternal and Child Health, Poznan University of Medical Sciences, 60-535 Poznan, Poland; 5Department of Immunobiology, Poznan University of Medical Sciences, 60-806 Poznań, Poland; 6Department of Gynecological Endocrinology, Jagiellonian University Medical College, 31-008 Cracow, Poland; 7Doctoral School, Poznan University of Medical Sciences, 61-701 Poznań, Poland

**Keywords:** epidemiology, HSIL, HPV testing

## Abstract

The prevalence and distribution of oncogenic human papillomavirus (HPV) genotypes in women who underwent screening for cervical cancer in the Wielkopolska region, Poland, were assessed, and the correlation of genotypes with the histological results was evaluated. Cervical samples were collected from 2969 women for cervical cancer screening. Participants were screened by liquid-based cytology and HPV genotyping (*n* = 1654) and referred to colposcopy and punch biopsy (*n* = 616) if recommended. HPV genotypes 16, 31, 52, 66, 53, and 51 are the most frequent types in the studied population. Genotypes 16 and 31 account for nearly one-fifth of the infections of diagnosed HPV infections. HPV 16, 31, and 52 are found in nearly 80% of premalignant HSIL lesions (CIN 2 and CIN 3). That leads to the conclusion that vaccination programs should cover as many types of HPV as possible and shows the urgent need to vaccinate the Polish population with a 9-valent vaccine.

## 1. Introduction

Human papillomavirus (HPV) plays a proven and undisputed role in cancer development among humans—cervical, anogenital, head and neck, and other locations [1]. The oncogenic potential of this virus was first suspected in the 1970s. Subsequent years brought more and more information on the different types of viruses, their quantity and characteristics, and the mechanisms responsible for initiating infection and cancer development [2,3,4,5,6]. Clinicians and researchers began to consider the most effective methods of population screening and looked forward to the vaccine, known as one of the most effective public health interventions [7]. Until now, more than 200 types of human papillomavirus have been fully sequenced [8,9,10]. Viruses typing allows HPV-based cervical cancer screening tests with some degree of genotyping [11,12], treatment of HPV infection, and application of HPV vaccines [13].

Infection caused by HPV is very common. Current data shows that more than 90% of the population is infected with HPV at some point in their lives, making this pathogen extremely prevalent in the general population [14]. In cervical infection, viruses enter the basal epithelial layer cells, causing the local infection, which is transient among many women. Persistent infection occurs among 10–20% of infected patients [15,16,17]. Some main factors and mechanisms protecting from the persistent infection and responsible for the “viral clearance” were found: the E7 peptide-specific CD4^+^ T-cells [18], mucosal and systemic (serum) antibodies against selected HPV antigens, local and systemic HPV-specific IgA and IgG antibodies, and vaccine-based antibodies (IgA, IgG) [19].

Thanks to the speedup of the screening, standardization, and improvement of the quality of the samples, liquid-based cytology (LBC) replaced the conventional Pap smear in developed countries [8,20]. In addition to the above-mentioned analysis of the collected cells, the sample taken in this way can be analyzed for the presence of HPV genetic material (by nucleic acid amplification with polymerase chain reaction (PCR) or signal amplification techniques), type indication, and its methylation degree [8,21,22,23]. More and more screening programs recommend the use of so-called solo HPV HR testing instead of LBC-based cytodiagnostics. According to current data presented, among others, in the Journal of the National Cancer Institute argues that the added sensitivity of the co-test to the single HPV test for the detection of curable cervical cancer applies to an extremely small number of women [24]. However, Poland is a country where the use of conventional cytology is rooted, and the high price of molecular diagnostics, unfortunately, reduces the frequency of its use in our conditions. So far, HPV testing is not included in the national screening program, more often used in private opportunistic screening.

In 2018, 570,000 new cervical cancer patients were reported globally, and 310,000 patients died [25]. Data from 2020 showed 604,000 new cases and 342,000 deaths worldwide. The authors of this analysis stated that cervical cancer was the most commonly diagnosed cancer in 23 countries and the leading cause of cancer death in 36 countries [26]. Data published in October 2021 by the Information Centre on HPV and Cancer indicated that in Poland, 3862 women are diagnosed with cervical cancer every year, and 2137 die from the disease. What is more, it is the third leading cause of cancer deaths in women aged 15 to 44 years in Poland, and 88.1% of invasive cervical cancers are attributed to HPV genotypes 16 or 18 [27]. This data is concerning regarding The Population-Based Cervical Cancer Early Detection Program, directed towards women aged 25–59 who have had no Pap-smear/LBC taken within the last two years and have never been treated for cervical cancer, financed by the National Health Fund [28]. It is difficult to estimate the incidence and mortality of cervical cancer in the Polish population in the coming years, as the percentage of women taking up screening opportunities fell dramatically during the COVID-19 pandemic [29].

In our study, we will answer the question about the epidemiology of HPV genotypes in the observed population. In addition, we are looking for the most sensitive and specific test when assessing the frequency in high-grade intraepithelial lesions. In the future, it may be possible to validate screening programs and plan the diagnosis of pre-cancerous changes of the cervix in the most favorable way, both for patients and in terms of the economy.

## 2. Materials and Methods

### 2.1. Study Design

We provide a prospective, ongoing 48-month, non-randomized pilot study in patients reporting to the Individual Specialised Medical Practise in 2018–2022. All subjects (*n* = 2969) were offered LBC and HPV genotyping tests, but it was their decision whether to perform them. All subjects from the study group underwent a verification diagnostic of abnormal LBC results, the suspicious clinical picture of the cervix, or the presence of an oncogenic genotype of HPV by punch biopsy. Abnormal LBC result means: ASC-US (atypical squamous cells of undetermined significance), AGC (atypical glandular cells), LSIL (low-grade squamous intraepithelial lesions), HSIL (high-grade squamous intraepithelial lesions), and cervical cancer whereas the visual assessment of the cervix was performed by an experienced colposcopist.

The Figure 1 presents the distribution of performed tests, LBC, HPV genotyping, and punch biopsy results. The Poznan University of Medical Sciences Bioethical Committee approved the study protocol (540/22). We included patients who met the following criteria: (i) aged over 18; (ii) non-pregnant subjects, postpartum; (iii) agreeing to the proposed surgical diagnostics in the case of indications and possible surgical treatment. The exclusion criteria were (i) the refusal of the possible treatment of SIL and (ii) a lack of technical possibility of performing the test.

### 2.2. Specimen Collection and Handling

#### 2.2.1. HPV Genotyping Test and LBC

We collected liquid-based cytology and molecular assessment samples with an endocervical Cyto-Brush preserved in PreservCyt ^®^ (Roche). Then, the probes were passed to an independent, standardized laboratory. PCR was performed, followed by a DNA enzyme immunoassay and genotyping with a reverse hybridization line probe assay for HPV detection. The lab technicians performed sequence analysis to characterize HPV-positive samples. The molecular test detected the DNA of 37 HPV genotypes (6, 11, 16, 18, 26, 31, 33, 35, 39, 40, 42, 45, 51, 52, 53, 54, 55, 56, 58, 59, 61, 62, 64, 66, 67, 68, 69, 70, 71, 72, 73, 81, 82, 83, 84, IS39, and CP6108).

#### 2.2.2. Colposcopy and Punch Biopsy

Further validation of abnormal screening results was performed on all patients with an abnormal smear above the ASCUS (as follows: ASC- US, AGC, LSIL, HSIL, cervical cancer), a positive HPV test for types 16, 18, 31, and a clinically suspicious cervical image. The Polish Society of Colposcopy and Cervical Pathophysiology recommended the International Federation of Cervical Pathology and Colposcopy classification.

### 2.3. Statistical Analysis

Analysis was conducted in SPSS, version 27, with the *p*-value set at 0.05. Dependencies between categorical variables were analyzed with the chi-square test. The sensitivity and specificity of different HPV types and cytology were analyzed using MedCalc statistical software.

## 3. Results

The study group included 1654 patients aged 18–86 (average age is 36). About 45% of the patients were nulliparous, and more than half had at least one child. The largest group was the group of patients aged 25–34 (675 subjects) and 35–44 (592 subjects). There were 242 women aged 45–54. The youngest and oldest groups were the least numerous and were represented by 76 and 65 women, respectively. HPV was detected in 781 subjects (47.2% of all patients). The most common types of HPV diagnosed among the sample were: 16, 31, 52, 66, 53, and 51. Table 1 shows the number of subjects diagnosed with those types of HPV and the percentage in the entire and HPV-positive groups. The most commonly diagnosed HPV type among the sample was type 16 (it was diagnosed among 14% of all patients and 30% of patients with any type of HPV, *n* = 233). The remaining distinguished types were observed among 67 to 85 subjects.

There was a statistically significant dependency between age and HPV type 16, 31, and 51. HPV 16 was observed more often among subjects aged between 25 and 34 (*p* < 0.001). HPV type 31 was more common among the youngest and oldest groups (*p* < 0.001). HPV 51 was observed among 9% of the youngest group, 4% of subjects aged 25–44, 2% of subjects aged 45–54; (*p* = 0.031), which presents in Table 2.

HPV 16, 31, and 51 were also related to biopsy results. Among all histopathological diagnoses, we observed: no pathology (246/616), koilocytosis—a microscopic image of HPV infection (46/616), LSIL (151/616), HSIL (166/616), and several diagnoses of cervical cancer and focal atypia. Due to the small number of patients diagnosed with focal atypia (3/616) and cervical cancer (4/616), we did not include the groups mentioned above in the statistical analysis. However, the calculations regarding the sensitivity and specificity of the test determining individual genotypes for detecting SIL changes were applied to all patients who underwent cervical biopsy (*n* = 616). Among subjects from HSIL and LSIL groups, there were greater proportions of patients with those types of HPV than among patients from the remaining groups (54% for HSIL and 23% for LSIL vs. 10% for no pathology and 11% for koilocytosis; *p* < 0.001 for HPV 16; 17% for HSIL and 9% for LSIL vs. 5% for no pathology and 2% for koilocytosis; *p* < 0.001 for HPV 31; 9% for HSIL and 8% for LSIL vs. 1% for no pathology and 2% for koilocytosis; *p* = 0.002 for HPV 51), as shown in Table 3.

The probability that HPV was positive when HSIL was positive (sensitivity) equaled: 54.22% for HPV 16, 16.87% for HPV 31, 7.83% for HPV 52, 5.42% for HPV 66, 4.22% for HPV 53, and 9.04% for HPV 51. Specificity of different types of HPV detecting HSIL equaled respectively: 85.33% for HPV 16, 94.22% for HPV 31, 93.78% for HPV 52, 94.00% for HPV 66, 94.22% for HPV 53, and 96.44% for HPV 51, as shown in Table 4.

The sensitivity of the different combinations of HPV types (ranged from 63.86% for HPV 16 and 31 to 79.52% for HPV 16, 31, 52, 66, 53, 51) was also increasing as more HPV types were added in the case of HSIL, and the specificity of those combinations was decreasing (ranged from 64.67% for HPV 16, 31, 52, 66, 53, 51 to 79.56% for HPV 16, 31). The lowest accuracy of detecting HSIL was observed for HPV 16, 31, 52, 66, 53 (68.18%), and the highest for HPV 16, 31 (75.32%), as shown in Table 5. The accuracy of detecting HSIL for HPV types separately ranged from 69.97% to 76.95%.

The probability that LBC results or HPV were positive when HSIL was positive (sensitivity) equaled: 91.57% for LBC and HPV 16, 85.54% for LBC and HPV 31, 82.53% for LBC and HPV 52, 83.13% for LBC and HPV 66, 83.13% for LBC and HPV 53, and 83.13% for LBC and HPV 51. Specificity of LBC results and different types of HPV detecting HSIL equaled respectively: 14.22% for LBC and HPV 16, 16.89% for LBC and HPV 31, 17.56% for LBC and HPV 52, 16.44% for LBC and HPV 66, 17.56% for LBC and HPV 53, and 17.56% for LBC and HPV 51, as shown in Table 6.

## 4. Discussion

Our study presents the prevalence of HPV in the Polish population in the Wielkopolskia Voivodeship and covers a group of 2969 subjects, from whom 1654 underwent HPV genotyping tests. The study showed the highest frequency of genotypes 16, 31, 52, 66, and 53 in the general population and 16, 31, and 52 in patients with histopathologically confirmed HSIL, respectively. There was a statistically significant dependency between age and HPV type 16, 31, and 51. HPV 16 was observed more often among subjects aged between 25 and 34 than among other groups. HPV type 31 was more common among the youngest (13%) and oldest group (9%) than among the rest of the subjects. HPV 51 was also the most common in the youngest group. These three HPV genotypes were also related to biopsy results. Among subjects from the HSIL group, there were a greater proportion of patients with those types of HPV than among patients from the remaining groups.

Modern knowledge and the rapid development of molecular testing techniques make it necessary to modify the current programs of both primary and secondary prevention. The impact of the implementation of HPV vaccination programs can already be seen in many European countries, and above all in Australia and New Zealand, where the highest decrease in the incidence of HPV-related diseases has been recorded. According to a 2018 publication by Patel et al., the use of the 9-valent vaccine might reduce the incidence of cervical cancer by 90% [30]. Our research group confirmed the high effectiveness of the 9-valent vaccine in the population of Polish women [25].

The incidence of certain highly oncogenic HPV genotypes may vary depending on demographics. For the purpose of the discussion, we have collected comparative data from the most extensive publications from different countries in Table 7. However, the data cannot be directly related because the studies were conducted on different groups of patients—some on the entire population and some only in histopathologically confirmed SIL.

What draws attention at the beginning is the prevalence of HPV genotype 16—it is the most common in each observed population, and the prevalence ranges from 3.9 to 63.7%. The data should be analyzed carefully because the study group in Russia was evaluated not based on histopathological diagnoses but only on cytology and included healthy women as well as those with pathology. Already data on the second most common HPV genotype are very divergent. The second most frequently observed type was 31 (in Spain, Canada, and France), which is consistent with the data obtained in our research group. However, in China, Russia, and New Zealand, the second most frequent genotype was 52, in Venezuela—HPV genotype 18, in Mexico—HPV genotype 51, and in Portugal—HPV genotype 58. The frequency of subsequent genotypes is even more divergent, which confirms the thesis that the above analysis has a cognitive value. Each latitude may differ in the frequency of occurrence of individual HPV genotypes and thus have a cervical cancer screening model based on other genotypes.

Not in all geographical areas, the proportion of histological HSIL+ in women infected with HPV 33 or HPV 31 was significantly different compared to women infected with HPV 16 (*p* = 0.30, *p* = 0.19, respectively) [40]. According to the data received and according to analyses from other countries, HPV 16 is highly oncogenic in the first place in terms of the frequency of occurrence. The percentage of HPV 18 in the population of infections and diagnosed lesions of the CIN 2 + type is decreasing. Other highly oncogenic types, including HPV 31, rank second in terms of frequency of occurrence in HSIL lesions.

This type of trend is also observed when comparing the results of meta-analyses from Poland. In the 2010 publication of women from Central and Western Poland diagnosed with HG SIL, the most common HPV genotypes were HPV 16, HPV 33, HPV 18, HPV 31, and HPV 56 [41]. Observation of epidemiological trends of HPV infection in countries with and without vaccination programs may necessitate the development of more than a 9-valent papillomavirus vaccine. Better primary prevention will affect the effectiveness of secondary prevention, which will ultimately reduce the costs associated with tertiary prevention and the treatment of oncological diseases related to HPV infection.

The recommended screening tests for HPV infection are tests detecting DNA of highly oncogenic HPV types with the possibility of genotyping or phenotyping. Material for molecular testing should be collected on certified, validated liquid media.

The obtained results indicate a new epidemiological distribution of human papillomaviruses, which only partially overlaps with global data and thus may be used in the future to better construct criteria for the diagnosis of squamous intraepithelial lesions. Going forward, perhaps as part of secondary prevention, this analysis and subsequent ones involving a wider population may contribute to the development of new vaccines against HPV.

## 5. Conclusions

HPV infection is one of the most common sexually transmitted infections in the study group; it was close to 47%. HPV genotypes 16, 31, 52, 66, 53, and 51 are the most common types. Genotypes 16 and 31 account for nearly one-fifth of the infections of diagnosed HPV infections. HPV 16, 31, and 52 are found in nearly 80% of premalignant HSIL lesions (CIN 2 and CIN 3). Vaccination programs should cover as many types of HPV as possible. The above study confirms the need to vaccinate the Polish population with a 9-valent vaccine.

## Figures and Tables

**Figure 1 pathogens-12-00350-f001:**
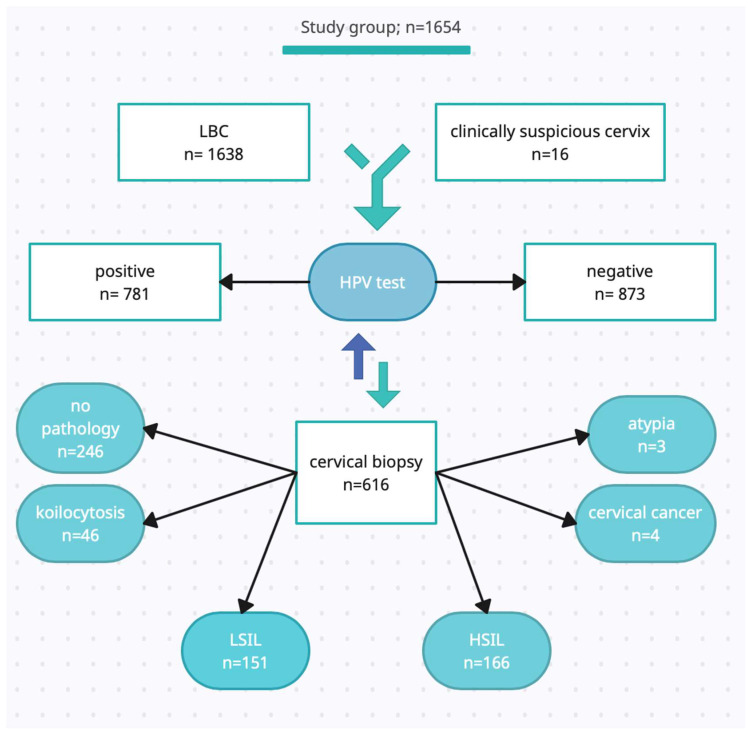
LBC—liquid-based cytology; HPV—human papillomavirus; LSIL—low-grade squamous intraepithelial lesions; HSIL—high-grade squamous intraepithelial lesions; *n*—number.

**Table 1 pathogens-12-00350-t001:** Frequency of HPV types among the group.

HPV Type	*n*	% of the Whole Group (*n* = 1654)	% of the HPV HPV-Positive (*n* = 781)
16	233	14.1	29.8
31	85	5.1	10.9
52	77	4.7	9.9
66	73	4.4	9.3
53	70	4.2	9.0
51	67	4.1	8.6

HPV—human papillomavirus; *n*—number.

**Table 2 pathogens-12-00350-t002:** Dependency between age and HPV types.

			Age			
HPV Type	<25 *n* = 75	25–34 *n* = 676	35–44 *n* = 592	45–54 *n* = 242	>=55 *n* = 65	*p*
	*n* (%)					
16	8 (10.5)	133 (19.7)	70 (11.8)	19 (7.9)	3 (4.6)	<0.001
31	10 (13.2)	43 (6.4)	18 (3.0)	7 (2.9)	6 (9.2)	<0.001
52	2 (2.6)	41 (6.1)	25 (4.2)	9 (3.7)	0 (0.0)	0.102
66	6 (7.9)	30 (4.4)	26 (4.4)	9 (3.7)	2 (3.1)	0.601
53	4 (5.3)	32 (4.7)	26 (4.4)	7 (2.9)	1 (1.5)	0.576
51	7 (9.2)	30 (4.4)	25 (4.2)	5 (2.1)	0 (0.0)	0.031

HPV—human papillomavirus; *n*—number; *p*—*p*-value for the chi-square test.

**Table 3 pathogens-12-00350-t003:** Dependency between biopsy results and HPV types.

HPV Type	No Pathology *n* = 246	Koilocytosis *n* = 46	HSIL *n* = 166	LSIL *n* = 151	*p*
	*n* (%)				
16	25 (10.2)	5 (10.9)	90 (54.2)	34 (22.5)	<0.001
31	12 (4.9)	1 (2.2)	28 (16.9)	13 (8.6)	<0.001
52	10 (4.1)	2 (4.3)	13 (7.8)	15 (9.9)	0.106
66	14 (5.7)	2 (4.3)	9 (5.4)	11 (7.3)	0.868
53	14 (5.7)	2 (4.3)	7 (4.2)	10 (6.6)	0.816
51	3 (1.2)	1 (2.2)	15 (9.0)	12 (7.9)	0.002

HPV—human papillomavirus; *n*—number; LSIL—low grade squamous intraepithelial lesions; HSIL—high grade squamous intraepithelial lesions; *p*—*p*-value for the chi-square test.

**Table 4 pathogens-12-00350-t004:** Sensitivity and specificity of different types of HPV in HSIL.

HPV Type	Histopathology Result		Total *n* = 616	Sensitivity	Specificity	PPV	NPV	Accuracy
	HSIL *n* = 166	Non-HSIL *n* = 450		Value (95% CI)				
16								
+	90	66	156	54.22% (46.32% to 61.96%)	85.33% (81.72% to 88.47%)	57.69% (51.18% to 63.95%)	83.48% (81.00% to 85.69%)	76.95% (73.42% to 80.22%)
-	76	384	460					
31								
+	28	26	54	16.87% (11.51% to 23.45%)	94.22% (91.65% to 96.19%)	51.85% (39.43% to 64.05%)	75.44% (74.08% to 76.76%)	73.38% (69.70% to 76.83%)
-	138	424	562					
52								
+	13	28	41	7.83% (4.24% to 13.02%)	93.78% (91.13% to 95.83%)	31.71% (19.77% to 46.66%)	73.39% (72.40% to 74.36%)	70.62% (66.85% to 74.19%)
-	153	422	575					
66								
+	9	27	36	5.42% (2.51% to 10.04%)	94.00% (91.39% to 96.01%)	25.00% (13.80% to 40.96%)	72.93% (72.07% to 73.78%)	70.13% (66.34% to 73.72%)
-	157	423	580					
53								
+	7	26	33	4.22% (1.71% to 8.50%)	94.22% (91.65% to 96.19%)	21.21% (10.64% to 37.83%)	72.73% (71.94% to 73.50%)	69.97% (66.18% to 73.57%)
-	159	424	583					
51								
+	15	16	31	9.04% (5.15% to 14.47%)	96.44% (94.29% to 97.95%)	48.39% (32.17% to 64.95%)	74.19% (73.20% to 75.15%)	72.89% (69.19% to 76.36%)
-	151	434	585					

“+” and “+”—true positive; “+” and “-”—false negative, “-” and “+”—false positive, “-” and “-”—true negative. HPV—human papillomavirus; *n*—number; LSIL—low-grade squamous intraepithelial lesions; PPV—positive predictive value; NPV—negative predictive value; 95% CI—95% confidence intervals.

**Table 5 pathogens-12-00350-t005:** Sensitivity and specificity of different combinations of HPV types in HSIL.

HPV Type	Histopathology Result		Total *n* = 616	Sensitivity	Specificity	PPV	NPV	Accuracy
	HSIL *n* = 166	Non-HSIL *n* = 450		Value (95% CI)				
16, 31								
+	106	92	198	63.86% (56.05% to 71.16%)	79.56% (75.53% to 83.19%)	53.54% (48.16% to 58.83%)	85.65% (82.90% to 88.01%)	75.32% (71.72% to 78.68%)
-	60	358	418					
16, 31, 52								
+	116	116	232	69.88% (62.29% to 76.75%)	74.22% (69.92% to 78.20%)	50.00% (45.37% to 54.63%)	86.98% (84.04% to 89.45%)	73.05% (69.36% to 76.52%)
-	50	334	384					
16, 31, 52, 66								
+	121	134	255	72.89% (65.46% to 79.49%)	70.22% (65.76% to 74.41%)	47.45% (43.25% to 51.69%)	87.53% (84.45% to 90.08%)	70.94% (67.18% to 74.50%)
-	45	316	361					
16, 31, 52, 66, 53								
+	122	152	274	73.49% (66.10% to 80.03%)	66.22% (61.65% to 70.58%)	44.53% (40.66% to 48.46%)	87.13% (83.90% to 89.80%)	68.18% (64.34% to 71.85%)
-	44	298	342					
16, 31, 52, 66, 53, 51								
+	132	159	291	79.52% (72.57% to 85.38%)	64.67% (60.05% to 69.09%)	45.36% (41.75% to 49.02%)	89.54% (86.29% to 92.09%)	68.67% (64.84% to 72.32%)
-	34	291	325					

“+” and “+”—true positive; “+” and “-”—false negative, “-” and “+”—false positive, “-” and “-”—true negative. PPV—positive predictive value; NPV—negative predictive value; 95% CI—95% confidence intervals.

**Table 6 pathogens-12-00350-t006:** Analysis of sensitivity and specificity of cytology and HPV—HSIL.

Cytology and HPV Type	Histopathology Result		Total *n* = 616	Sensitivity	Specificity	PPV	NPV	Accuracy
	HSIL *n* = 166	Non-HSIL *n* = 450		Value (95% CI)				
16								
+	152	386	538	91.57% (86.25% to 95.31%)	14.22% (11.13% to 17.80%)	28.25% (27.06% to 29.48%)	82.05% (72.50% to 88.80%)	35.06% (31.29% to 38.98%)
-	14	64	78					
31								
+	142	374	516	85.54% (79.26% to 90.51%)	16.89% (13.54% to 20.68%)	27.52% (26.05% to 29.04%)	76.00% (67.47% to 82.86%)	35.39% (31.61% to 39.31%)
-	24	76	100					
52								
+	137	371	508	82.53% (75.88% to 87.98%)	17.56% (14.15% to 21.39%)	26.97% (25.39% to 28.61%)	73.15% (64.92% to 80.04%)	35.06% (31.29% to 38.98%)
-	29	79	108					
66								
+	138	376	514	83.13% (76.55% to 88.49%)	16.44% (13.14% to 20.20%)	26.85% (25.31% to 28.45%)	72.55% (63.99% to 79.72%)	34.42% (30.67% to 38.32%)
-	28	74	102					
53								
+	138	371	509	83.13% (76.55% to 88.49%)	17.56% (14.15% to 21.39%)	27.11% (25.55% to 28.74%)	73.83% (65.58% to 80.69%)	35.23% (31.45% to 39.14%)
-	28	79	107					
51								
+	138	371	509	83.13% (76.55% to 88.49%)	17.56% (14.15% to 21.39%)	27.11% (25.55% to 28.74%)	73.83% (65.58% to 80.69%)	35.23% (31.45% to 39.14%)
-	28	79	107					

“+” and “+”—true positive; “+” and “-”—false negative, “-” and “+”—false positive, “-” and “-”—true negative. PPV—positive predictive value; NPV—negative predictive value; 95% CI—95% confidence intervals.

**Table 7 pathogens-12-00350-t007:** The incidence of oncogenic HPV genotypes in different regions.

Country	*n*	Positive	Group	16	18	31	39	45	51	52	58	59
Mexico [31]	129	38%	A	15	0.8	3.1	3.1	1.6	3.9	3.1	2.3	2.3
Spain [32]	533	47%	B	35.3	6.8	14.4	1.7	3.6	3.9	7.7	10.9	2.8
Canada [33]	238	100%	C	63.7	8.3	10.4	3.0	0.3	0.3	4.3	1.3	0.1
Venezuela [34]	142	100%	C	63.3	9.8	7.5	-	4.2	3.5	4.9	4.9	-
China [35]	641	53.4%	B	14.4	8.0	4.8	0.3	1.4	0.3	10.8	4.8	1.7
New Zeland [36]	362	94%	D	50.8	12.1	17.1	7.3	4.5	10.1	18.8	11.5	4.2
Russia [37]	841	13%	E	3.9	0.5	2.8	0.4	0.7	0.6	1.7	0.5	0.4
Portugual [38]	582	98%	B	58	3.5	10.4	0.4	0.9	3.5	5.1	7.7	1.1
France [39]	493	98%	D	62.3	4.3	15.4	2.4	1.2	7.7	8.7	6.5	0.2

Group A—histopathologically with or without SIL (CIN 1,2,3); group B—histopathologically with or without SIL (CIN 1,2,3, cervical cancer); group C—histopathologically with CIN 3; group D—histopathologically with CIN 2 or 3; group E—cytological diagnosis (from NILM to cervical cancer).

## Data Availability

All data available at D. Pruski, dominik.pruski@icloud.com.

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
