# Peer review of "Prevalence of HPV and Assessing Type-Specific HPV Testing in Cervical High-Grade Squamous Intraepithelial Lesions in Poland"

_pathogens, 2023, doi:10.3390/pathogens12020350_

Round 1

Reviewer 1 Report

The article "Prevalence of HPV and assessing type-specific HPV testing in cervical high grade squamous intraepithelial lesions in Poland", I find very interesting, they analyze the prevalence of HPV infection in Poland by virus detection, not only assessing the prevalence but also the frequency about age ranges, the most frequent genotypes in both LBC and biopsies to highlight the importance of vaccination. However, I have some observations.

I suggest reviewing the figures, sorted by age of the 1654 patients.

In the image showing the prevalence distribution diagram, I suggest checking the spelling.

In table 1 there is an error with genotype 5 instead of 51.

The results in table 2 and those described in rows 134-141 are redundant, I suggest that only the significant ones are described.

I find it surprising that there is no data for HPV18, being one of the most frequent genotypes, nor is there any mention of the possibility of diagnosing multi-infection.

I would have liked to see a representative picture of the diagnosis of the 37 HPV genotypes reported in the methodology.

It is true that prevalences of HPV genotypes vary in different regions, is there any evidence to suggest the absence of genotype 18 in the population? And in the discussion, could you include information on how vaccination strategies have impacted these prevalences?

The sensitivity and specificity analyses they present for HSIL should be better discussed in the article.

Please check the format of references according to the journal, as they are not written homogeneously.

Finally, I think there should be a section on the limitations of the study.

Author Response

Please read the attached file.

Reviewer 2 Report

The manuscript by Marcin and colleagues assessed the prevalence of HPV and evaluated the correlation with histological results in women with screening. They identified 5 most frequent types. The study provided an exploration of HPV prevalent in women in Poland.

Major points:

1.       Does the author detect HPV types in addition to the 5 types (16,31,52,66,53,5)? If yes, it would be helpful to provide data on those types. Were there any samples infected with multiple HPV types?

2.       Page 3 line 102-108: Provide a list of 37 types detected by the assay. Why were samples HPV positive but with unknown genotypes? Detailed information of the testing needs to be provided. Are there any data for the samples been sequenced?

3.       The manuscript looks somewhat chaotic and some formal requirements for the manuscript are not met: for example, the author contribution part is incomplete, title of table 7 is missing.

Minor points:

1.       Page 1 line 39-40: references were not inserted correctly.

2.       Page 2 line 62, insert period before “So far”.

3.       Page 2 line 64, insert a comma after 2008.

4.       Page 2 line 84-85, How were “abnormal LBC results” and “suspicious clinical picture of the cervix” defined?  

5.       Page 2 line 88: what does "(XXXX/XX)" mean?

6.       Page 3, Provide a figure legend for the flow chart.

7.       Page section 22.1, The total subject in the study is 2969, but there were 1654 patients in study group. I assume 1969 subject met the selection criteria. If this is right, please provide this information in the text.

8.       Page 3 line 94-99, this content should be moved to introduction.

9.       Page 4 line 141: “what presents in table 2” is not clear, rephrase.

10.   Page 6 line 170-172: the content should be footnote for table 4.

11.   Page 7 line 183: the content should be footnote of table 5.

Author Response

Please read the attached file.

Round 2

Reviewer 1 Report

I think the article is much improved in form and substance, and I consider it now adequate, maybe the editorial design could correct the format of the tables

Author Response

Thank you very much for your valuable comments. 

Reviewer 2 Report

1. The total subject in the study is 2969, but there were 1654 patients in study group. I assume 1969 subject met the selection criteria. If this is right, please provide this information in the text.

Author response: Two thousand nine hundred sixty-nine patients is a whole group of patients who were offered the HPV genotyping test, but it was their decision whether to perform it.

This information should be in the manuscript

 2. Page 3 line 98: what does "(XXXX/XX)" mean?

Author response: The number of Bioethical Cometee agreement was added, thank you for noticing it.

The number was not added.

3. Author contribution should be specific

4. Page 3 line 112, “The lab technicians performed sequence analysis to characterise HPV-positive samples”. was the sequencing data related to this study? if not I suggest to remove it.

5. check tables and make sure they arr in same format

Author Response

Thank you. Please read the attached file.
